# Common hematological parameters reference intervals for apparently healthy pregnant and non-pregnant women of South Wollo Zone, Amhara Regional State, Northeast Ethiopia

**Mesfin Fiseha** [1]*, **Miftah Mohammed** [1], **Endris Ebrahim**[1], **Wondmagegn Demsiss**[1], **Mohammed Tarekegn**[2], **Amanuel Angelo**[3], **Mikias Negash** [4], **Zemenu Tamir**[4], **Mihret Tilahun**[1], **Aster Tsegaye** [4]

1 Department of Medical Laboratory Science, College of Medicine and Health Sciences, Wollo University, Dessie, Ethiopia, 2 Kombolcha 03 Health Center, Kombolcha, South Wollo, Amhara Regional State, Ethiopia, 3 Medical Laboratory Department, Saint Peter Specialized Hospital, Addis Ababa, Ethiopia, 4 Department of Medical Laboratory Sciences, College of Health Sciences, Addis Ababa University, Addis Ababa, Ethiopia

* mesfinfiseha40@gmail.com

**Data Availability Statement:** All the relevant data are within the Manuscript.

## Abstract

### Background

Hematological reference intervals (RIs) are affected by inherent variables like age, sex, genetic background, environment, diet and certain circumstances such as pregnancy signifying the need for population specific values. This study was designed to establish RIs for common hematological parameters of apparently healthy pregnant and non-pregnant women from Northeast Ethiopia.

### Method

This community based cross-sectional study recruited 600 pregnant and non-pregnant women in South Wollo Zone, Northeast Ethiopia from June to August 2019. Complete blood count was performed for eligible participants using Mindary BC-3000 plus hematology analyzer. The mean, median, and 2.5th and 97.5th percentile reference limits with 90% CI were determined using SPSS version 23.

### Result

The established selected 2.5th–97.5th percentiles RIs for pregnant women were: WBC: 4.0–13.2x10$^9$/L; RBC: 3.45–4.67x10$^{12}$/L; Hgb: 10.1–13.7g/dL; HCT: 33.5–46.5%; MCV: 85-104fL; MCH: 27.5–33.0pg; MCHC: 30.3–33.7g/dL and Platelet count: 132-373x10$^9$/L. The respective values for non-pregnant women were 3.6–10.3; 4.44–5.01; 12.4–14.3; 38.4–50.1; 86–102; 27.1–32.4, 30.4–34.1, 173–456. A statistically significant difference between pregnant and non-pregnant women was noted in all hematological parameters except MCHC. The mean and median value of WBC count, MCV, MPV and PDW increased, whereas mean values of HCT and Platelet count decreased as gestational period advances.

**Funding:** The Author (s) received specific funding for this work from Addis Ababa University and the Ministry of Science and Technology. However, the funders had no role in study design, data collection and analysis, and preparation of the manuscript.

**Competing interests:** The authors have declared that no competing interests exist.

## Conclusion

The observed difference from other studies signify the necessity for using trimester specific RIs and separate RI for pregnant and non-pregnant women. Thus, we recommend the health facilities in the study area to utilize the currently established RIs for pregnant and non-pregnant women for better care.

## Background

Hematological parameters are among the most commonly requested laboratory tests that assist health care professionals to make a suitable evidence-based diagnoses and therapeutic judgments for their patients [1,2]. These tests are valuable in diagnosing anemia, certain cancers, infection, acute hemorrhagic states, allergies, and immunodeficiency. Moreover, the tests are also important for monitoring of side effects of certain drugs, response to treatment, and determining the effects of chemotherapy and radiation therapy on blood cell production and recruitment of eligible participants for clinical trials including vaccines [3,4].

Reference Intervals (RIs) are the critical element and common decision support tool for accurate and reliable interpretation of hematological test results. [5–7]. They can be affected by several pre-analytical, analytical, pathological, and physiological factors such as the technique and time of blood collection, transport, and storage of the specimens, posture, exercise and methodology, and instrument used to obtain the result. Likewise, inherent variables like age, sex, genetic background, environment, diet, and even certain circumstances such as pregnancy also affect RIs. Moreover, it can also be affected by population and ecological factors such as ethnic background, climate, and altitude [8–12].

Maternal mortality and morbidity continue to be a substantial problem in low-income countries, despite a worldwide focus on the need to improve maternal health as part of attaining sustainable development goals [13]. One of the utmost underlying causes of maternal mortality is hematological complications like anemia and thrombocytopenia [14–16]. This is because pregnancy places intense stress on the hematological system and several quantitative changes happen [17,18]. These complications and physiological changes observed in pregnant women need accurate and reliable hematological parameter RIs [19–21]

Even if there is a common pregnancy-induced laboratory abnormality (mainly anemia and thrombocytopenia) and significant variations between women and men are well documented, very few laboratories routinely provide reference values for pregnant and non-pregnant women [8,22–26]. Indeed, many laboratories in Ethiopia have been using RIs established from non-Ethiopian participants, mostly from western countries which are available in textbooks, inserted leaflet or manufacturer manual [26].

For instance, a study conducted in west Kenya and China found that utilizing adult female RIs for pregnant women resulted in the greatest misclassification of pregnant women. In addition, applying non-local adult female Hgb value RIs for pregnant women resulted in a high proportion of out of range values [27,28]. Failure to use their own locally derived RIs may lead to unnecessary therapeutic actions without determining the real cause of the abnormality, increases the risk of overlooking important physiologic alterations resulting from pathological conditions and of misinterpreting normal changes as pathological events, incorrect clinical care for the peoples, unnecessary exclusion of eligible participants as well as over-reporting of adverse events in clinical trials [8,27–30].

Thus, this study was designed to determine common hematological parameters RIs for apparently healthy pregnant and non-pregnant women that would be utilized in South Wollo Zone, Amhara Regional State, Northeast Ethiopia.

## Materials and methods

### Study design, area and setting

A community based cross sectional study was done in South Wollo Zone, Amhara Regional State, Northeast Ethiopia from June to August 2019. South Wollo Zone is located in the North East part of Ethiopia lying between latitudes from 10˚10'N to 38˚ 28'E longitudes. Its zonal capital, Dessie, is found 400 Km of Addis Ababa. The elevation ranged from the dry plains at 1,000 m altitude in the east to the high peaks above 3,500 m altitude in the west. However, highland areas ranging between 1,500 and 3,500 m altitude are the dominating features of South Wollo Zone. According to the national population and housing census of Ethiopia, the projected population of the zone for 2017 was estimated to be 3,087,132 (525,771 in urban and 2,561,373 in rural) [31,32].

### Population

Apparently healthy pregnant and non-pregnant women with a reproductive age group (15–49 years) who lived in South Wollo Zone for 5 years and fulfilling the eligibility criteria were included in the study. Moreover, willingness of individuals demonstrated by completion and signing/Thumb printing of the consent form and willingness to provide the biological samples required. Both *a priori* and *posteriori* selection methods were applied. Participants were excluded from the study if one of the following criteria is fulfilled:

- Individuals with history of chronic diseases, blood transfusion in the last one year, blood donation in the last three months, hospital admission for the last 1 year, surgical procedure in the last three years, chronic gastritis, malaria in the last 6-month, tuberculosis in the last two years, cancer, cardiac illness, bleeding disorder, allergy, wheezing, liver disease and thyroid disease.

- Individuals who had febrile symptoms and high blood pressure ($>$ 140 mmHg for diastolic blood pressure and $>$ 90 mmHg for systolic blood pressure).

- Individuals taking all prescription drugs (except iron/folic acid for pregnant women since it is routinely given) and had frequent habit of smoking, chewing *Khat* and alcohol consumption.

- Pregnant women who had active bleeding and obstetrics complications

- Subjects performing exercise/physical training prior to blood collection.

- Obese individuals (Body mass index greater than or equal to 30 kg/m$^2$) [33].

- Subjects positive for human immunodeficiency virus (HIV), Hepatitis B surface antigen (HBsAg), Hepatitis C virus (HCV), Treponemal (TP) Ab, C-reactive protein (CRP), hemoparasite and intestinal parasite infection.

- Individuals with frequent exposure for hazardous chemical like gas station and factory workers.

- Participants with poor quality of Blood specimens (Hemolyzed, Lipemic and Clotted)

- In addition, for non-pregnant women, those who were menstruating during data collection and breast-feeding, and women taking oral contraceptive were excluded.

## Sample size determination and sampling techniques

The sample size was determined based on IFCC and CLSI (2000) guideline which recommends a minimum sample size of 120 per partition by using non-parametric method with a power of 90% [12]. This study requires four partitions (three trimesters for pregnant women and one for non-pregnant women). Therefore, four hundred eighty study participants were needed (4x120 = 480). According to previous study conducted in western Kenya, about 20% of study subjects [27] were not qualifying for the final reference interval determination due to various reasons like HIV infection etc. Considering a 20% exclusion or dropout from data analysis, the sample size was corrected as follows [34]:

$$\text{Corrected sample size} = \text{Original sample size} * (1/(1-\text{exclusion rate}))$$
$$= 480 * (1/(1-0.20) = 600$$

The total sample size for the study was 600 participants, with 150 individuals per each partition or subgroup.

The study participants were recruited first, by considering altitude (lowland and highland) and residence (urban and rural) differences; four woredas were selected purposively from the study area: Dessie Town, Kombolcha Town, Kalu woreda and Kutaber woreda. Secondly, the determined sample size was allocated for each selected woredas per partition proportional to their population size. Thirdly, four kebeles (smallest administration units) each from the selected woredas were identified conveniently as long as study areas were easy to reach and suitable for biological sample transportation to the laboratory. Finally, the reference individuals were recruited by convenient sampling techniques from each selected kebele with the respective partition until the required sample size was attained. While recruiting study participants, one individual per household per partition was included in the study.

## Data collection procedure

All potential participants feeling subjectively well were recruited by health extension workers from the community in the selected kebels, and invited to come to the central data collection center. Initially, the study participants agreed to give written consent and assent after being informed about the purpose of the study and its associated risks. Then participants underwent a physical examination by an experienced clinician. Socio-demographic, clinical data, and other important information was collected using a pre-tested interviewer guided structured questionnaire adopted from the CLSI guidelines and other related literature. Finally, eligible participants provided biological specimens like blood, urine, and stool specimens for screening of intestinal parasites, HIV, HBsAg, anti-HCV, Syphilis, pregnancy, and measurement of hematological parameters following standard operating procedures (SOPs).

## Laboratory analysis

**Specimen collection, handling, transportation and storage and processing.** After completion of the interview, about 4 ml of venous blood was taken by an experienced laboratory technologist from each study participant in Ethylene Diamine Tetra Acetic Acid (EDTA) Vacutainer tubes (Becton-Dickinson, Franklin Lakes, New Jersey, USA) for hematological and serological tests. Moreover, a peripheral blood film was prepared to detect morphological abnormalities in blood cells and hemoparasite. Venous blood specimens were mixed

thoroughly by gently inverting about eight times. The test tube was appropriately labelled, placed in a vaccine carrier, and transported to Dessie health center laboratory within 3 hours of collection at room temperature for analysis. To minimize diurnal variation, specimens were drawn in the morning between 8.00 and 11.00 am and processed within 8 hours of collection.

Stool specimens were collected with a clean container for the detection and identification of intestinal parasites. For the urine pregnancy test, all enrolled participants provided approximately 15ml of urine specimen in a clean, screw-capped container, and it was processed immediately. All the above specimen collection and processing procedures adhere to good clinical laboratory practices and follow established SOPs.

**Screening tests.** By using plasma specimen infectious disease screening was performed for HIV-I and II (HIV 1/2 STAT-PAK: Chembio Diagnostic systems, INC. Medford, New York, USA; SD BIOLINE HIV 1/2 3.0: SD standard diagnostics, INC. Republic of Korea; ABON HIV 1/2/0 tri-line: Abon Biopharma Co., Ltd. Hangzhou, China.), HBsAg (ACON Biotech, INC, Co., Ltd. Hangzhou, China), HCV (ACON Biotech, INC, Co., Ltd. Hangzhou, China) and Syphilis (ACON Biotech, INC, Co., Ltd. Hangzhou, China) by rapid test Kits at Kombolcha 03 health center. A rapid AVITEX CRP (Omega Diagnostics LTD, Scotland, USA) latex agglutination test kit was used for measurement of CRP in human serum or plasma. HCG tests for pregnancy were performed by using the Laboquick pregnancy test strip (Labex Engineering, Ltd., Bulgaria). This test is used for qualitative determination of HCG in urine specimens for early detection of pregnancy. Direct stool microscopic examination using saline was performed to detect and identify intestinal parasites. Thick and thin blood smears were prepared and stained with Giemsa, and examined microscopically for hemoparasites and blood cell abnormalities.

**Hematological parameter analysis.** Hematological parameters were measured by Mindray BC-3000 plus, an automated 3-part differential hematology analyzer (Mindray Corporation, Kobe, China), that generates 19 parameters at a time (WBC, Lymph#, MID#, Gran#, Lymph%, MID%, Gran%, RBC, Hgb, HCT, MCV, MCH, MCHC, RDW-CV, RDW-SD, PLT, MPV, PDW, PCT). The machine utilized two basic principles: the impedance method for determining the WBC, RBC, and PLT and the colorimetric method for determining Hgb.

**Quality assurance.** To ensure the quality of the data, the questionnaire was translated to Amharic and checked for its consistency through back translation into English by different individuals. Two days training was provided for the data collectors (Health officer and lab technologists) and community mobilizers (Health extension workers) about the objective of the study, study participants' rights, confidentiality of patient information, procedure of physical examination, specimen collection and measurements, and how to approach and interview participants before the actual data collection by the principal investigator and experienced clinician.

A protocol for sample collection, processing, transportation and storage was strictly followed to have safe procedures and reliable specimens. Prior to analysis, each specimen and questionnaire were evaluated for acceptability and rejection criteria. The performance of the instrument and reagents was controlled by running quality control daily prior to the start of the test. As an internal quality assurance, commercial or in-house quality control specimens were run daily and in every batch. For hematological tests, the three levels of commercial controls (Low, Normal and High) were run daily after mixing well by inverting 8–10 times. Test specimens were run when the quality control result was accepted. Cross checking of the proper verification, recording and entering the result in laboratory result form and in the software was carried out. Moreover, our laboratory is enrolled in the one world accuracy proficiency program for hematological test analysis, and it had satisfactory performance for all hematological parameters processed over the study period.

### Data analysis, interpretation and presentation

Once the data was cleaned, edited, and checked for completeness, it was entered into the Statistical package for social science version 23 (SPSS version 23.0, SPSS Inc. Chicago, IL, USA) for statistical analysis. The data was tested for normality of its distribution by the Kolmogorov-Sminro test. The Dixon and Reed method was used to find outliers within each subgroup and parameter. The CLSI/IFCC guidelines were followed in determining the RIs, which were 2.5% and 97.5% in a ranked list of reference value data using the formula: the lower limit has the rank number 0.025 (n+1) and the upper limit has the rank number 0.975 (n+1) [12]. Out of range (OOR) scores were also produced by comparing the current study's findings with company-derived values to assess how much misclassification occurred. Because the majority of the hematological parameters had a non-normal distribution, trimester-specific variations of hematological parameters were assessed using the Krusal-Wallis test, and pregnant and non-pregnant hematological parameter variations were assessed using the Mann-Whitney U test. A two-sided p-value of <0.05 was considered statistically significant.

### Ethical considerations

The study protocol was ethically approved by the research and ethics review committee of the Department of Medical Laboratory Sciences, College of Health Sciences, Addis Ababa University. A letter of permission to conduct the study was obtained from the South Wollo Zonal Health department, the woreda health office and the respective kebeles catchment health facilities. Written informed consent was sought after the study participants were informed about the aim, risks, and benefits of the study and prior to involving them in the study. Confidentiality was kept and interruption was entertained at any stage of the study. Personal identifiers were not used in the questionnaire and in the lab analysis; rather, unique identifier was employed. Individuals positive for infections and other disease conditions were linked to the nearby government health facilities for further counseling, diagnosis and treatment accordingly. Study participants were adequately counseled before HIV testing and done by a trained counselor.

## Result

### Socio demographic and clinical characteristics of the study participants

A total of 600 participants aged 15 to 49 years were enrolled in the study. Of these, 450 were pregnant and 150 were non-pregnant women. The average age of pregnant and non-pregnant women was 26.1 years and 25.8 years, respectively. The majority of the participants, both pregnant and non-pregnant women, were urban dwellers (52.0%) and lived in lowland areas (54.7%) (Table 1).

Of the 600 participants, 533 (88.8%) fulfilled all of the eligibility criteria for the establishment of reference intervals and distributed as 136 women in the 1st, 130 in the 2nd, 131 in the 3rd trimester of pregnancy and 136 non-pregnant women. The remaining 67 (11.2%), 20 (3.33%) in the second trimester, 19 (3.17%) in the third trimester, 14 (2.33% %) in the first trimester, and 14 (2.33% %) non-pregnant women were excluded for the following reasons: positive serological tests (HIV, HBsAg, HCV, Syphilis and CRP), history of chronic diseases, recent blood transfusion and donation, obesity, and high blood pressure. Outlier data was likewise removed from the final RIs calculation resulting in different sample size for each parameter.

### Hematological parameter RIs in pregnant and non-pregnant women

The mean, median and 2.5$^{th}$ -97.5$^{th}$ percentile RIs of common hematological parameters of pregnant women are summarized in Table 2. As shown in the table, all white blood cell

**Table 1. Socio demographic characteristics of study participants in South Wollo zone, Amhara Region, Northeast Ethiopia from June to August, 2019.**

| Variables | Category | Pregnant women | | Non-pregnant women | |
|---|---|---|---|---|---|
| | | Frequency (N = 450) | Percentage (%) | Frequency (N = 150) | Percentage (%) |
| **Woredas** | Dessie | 150 | 33.3 | 54 | 36.0 |
| | Kutaber | 72 | 16.0 | 22 | 14.6 |
| | Kombolcha | 84 | 18.7 | 25 | 16.7 |
| | Kalu | 144 | 32.0 | 49 | 32.7 |
| **Educational Status** | Illiterate | 42 | 9.3 | 3 | 2.0 |
| | Read and write | 15 | 3.3 | 0 | 0.0 |
| | Primary | 174 | 38.7 | 12 | 8.0 |
| | Secondary | 143 | 31.8 | 16 | 10.7 |
| | Diploma and above | 76 | 16.9 | 119 | 79.3 |
| **Occupation** | Student | 9 | 2.0 | 57 | 38.0 |
| | House wife | 314 | 69.8 | 15 | 10.0 |
| | Government employee | 55 | 12.2 | 70 | 46.7 |
| | Private employee | 31 | 6.9 | 6 | 4.0 |
| | Farmer | 16 | 3.5 | 2 | 1.3 |
| | Merchant | 25 | 5.6 | 0 | 0.0 |
| **Marital status** | Single | 10 | 2.2 | 81 | 54.0 |
| | Married | 438 | 97.4 | 62 | 41.3 |
| | Divorced | 2 | 0.3 | 6 | 4.0 |
| | Widowed | 0 | 0.0 | 1 | 0.7 |
| **Religion** | Orthodox Christian | 143 | 31.8 | 74 | 49.3 |
| | Muslim | 303 | 67.3 | 76 | 50.7 |
| | Protestant | 4 | 0.9 | 0 | 0 |
| | Others | 0 | 0.0 | 0 | 0 |
| **Residence** | Rural | 216 | 48.0 | 72 | 48.0 |
| | Urban | 234 | 52.0 | 78 | 52.0 |
| **Altitude** | Lowland | 246 | 54.7 | 82 | 54.7 |
| | Highland | 204 | 45.3 | 68 | 45.3 |

parameters (WBC, Lymph #, MID #, Gran #, Lymph %, MID %, and Gran %) showed statistically significant differences (P-value< 0.05) between pregnant and non-pregnant women. Pregnant women had a higher median value of WBC, Gran #, Gran%, MID% and MID# than their counterpart. Whereas non-pregnant women had higher median values for Lymph # and Lymph% than pregnant women. Pregnant women had significantly lower RBC count, Hgb, HCT, PLT, PCT and higher MCV, MCH, RDW-CV, RDW-SD, MPV, PDW than non-pregnant women; however there was no statistically significant difference in MCHC.

**Hematological parameter RIs in pregnant women according to trimester.** Table 3 displays the mean, median, RI (2.5th-97.5th), 90% CI of upper and lower reference limits, and P-values for comparison between trimesters; that is, 1st vs. 2nd, 1st vs. 3rd, and 2nd vs. 3rd trimester of pregnant women. The median values of WBC, Gran #, and Gran% increased as the gestational period moved from 1st to 3rd trimester. The median values of lymph# and lymph% decreased in the 2nd trimester and increased in the 3rd trimester.

RBC count and Hgb levels were higher in the 1st trimester (3.86x10$^{12}$/L and 12.3g/dL), lower in the 2nd trimester (3.49x10$^{12}$/L and 11.6g/dL), and higher in the 3rd trimester (3.78x10$^{12}$/L and 12.7g/dL). However, the median values of HCT decreased as gestational age increased. There was no statistically significant difference in MCH, MCHC, RDW-CV, and RDW-SD between trimesters, 1st vs. 2nd, 1st vs. 3rd, and 2nd vs. 3rd. On the other hand, MCV showed statistically significant variations between trimesters and its mean and median values increased as the gestational period advanced.

Platelet count demonstrated a statistically significant difference between trimesters, 1st vs. 2nd, 1st vs. 3rd, and 2nd vs. 3rd trimester of pregnant women, as shown in Table 3. Between

**Table 2. Mean, median, 95% (2.5th- 97.5th) RIs with 90% CI for lower and upper reference limits of hematological parameters of pregnant and non-pregnant women of South Wollo zone, Amhara Region, Northeast Ethiopia from June to August, 2019.**

| Parameter | | N | Mean | Median | RI (2.5th-97.5th) | 90% CI | | P-value |
|---|---|---|---|---|---|---|---|---|
| | | | | | | LLC | ULC | |
| WBC (x10⁹/L) | Pregnant | 395 | 8.3 | 8.1 | 4.0–13.2 | 3.7–4.4 | 12.7–15.1 | 0.0001* |
| | Non- pregnant | 136 | 6.6 | 6.5 | 3.6–10.3 | 3.5–3.9 | 9.7–11.1 | |
| Lymph#(x10⁹/L) | Pregnant | 395 | 1.9 | 1.9 | 1.1–2.71 | 1.0–1.1 | 2.7–3.0 | 0.0001* |
| | Non-pregnant | 136 | 2.3 | 2.3 | 1.24–3.7 | 1.1–1.4 | 3.4–3.7 | |
| MID# (x10⁹/L) | Pregnant | 393 | 0.6 | 0.5 | 0.2–1.0 | 0.2–0.3 | 1.0–1.1 | 0.020* |
| | Non-pregnant | 135 | 0.5 | 0.4 | 0.2–0.8 | 0.1–0.3 | 0.7–0.8 | |
| GRAN# (x10⁹/L) | Pregnant | 393 | 5.8 | 5.8 | 2.2–9.8 | 1.9–2.4 | 9.6–11.6 | 0.0001* |
| | Non-pregnant | 135 | 3.7 | 3.5 | 1.3–6.9 | 1.0–1.6 | 6.2–7.2 | |
| Lymph (%) | Pregnant | 384 | 23.4 | 22.5 | 12.9–38.1 | 11.7–13.4 | 37.2–41.4 | 0.0001* |
| | Non-pregnant | 134 | 36.6 | 35.7 | 19.9–57.1 | 13.9–23.3 | 54.1–62.1 | |
| MID (%) | Pregnant | 390 | 7.6 | 7.4 | 4.2–11.6 | 3.9–4.7 | 11.2–12.9 | 0.002* |
| | Non-pregnant | 135 | 6.9 | 6.8 | 3.9–10.9 | 3.7–4.0 | 10.7–12.0 | |
| GRAN (%) | Pregnant | 390 | 69.3 | 70.4 | 50.5–81.5 | 49.3–52.8 | 81.2–84.9 | 0.0001* |
| | Non-pregnant | 135 | 55.5 | 56.1 | 33.4–71.8 | 27.7–38.0 | 70.1–75.5 | |
| HGB (g/dL) | Pregnant | 394 | 10.9 | 11.9 | 10.1–13.7 | 9.9–10.4 | 13.31–13.98 | 0.0001* |
| | Non-pregnant | 135 | 13.8 | 13.1 | 12.4–14.3 | 11.8–12.6 | 14.1–14.6 | |
| RBC (x10¹²/L) | Pregnant | 396 | 3.5 | 3.52 | 3.45–4.67 | 3.36–3.52 | 4.45–4.98 | 0.0001* |
| | Non-pregnant | 134 | 4.7 | 4.76 | 4.44–5.01 | 4.21–4.83 | 4.91–5.23 | |
| HCT (%) | Pregnant | 396 | 40.2 | 40.4 | 33.5–46.5 | 32.3–34.0 | 46.1–49.7 | 0.0001* |
| | Non-pregnant | 134 | 44.2 | 44.6 | 38.4–50.1 | 37.3–39.1 | 48.8–50.9 | |
| MCV (fL) | Pregnant | 391 | 94.7 | 94.7 | 84.8–103.5 | 83.7–87.2 | 103.0–106.6 | 0.001* |
| | Non-pregnant | 135 | 92.3 | 92.6 | 86.1–101.6 | 83.5–86.7 | 99.4–102.8 | |
| MCH (pg) | Pregnant | 370 | 30.3 | 30.2 | 27.5–33.0 | 27.2–27.7 | 32.8–33.9 | 0.036* |
| | Non-pregnant | 133 | 29.7 | 29.8 | 27.1–32.4 | 26.1–27.4 | 31.9–33.3 | |
| MCHC (g/dL) | Pregnant | 389 | 31.9 | 31.9 | 30.3–33.7 | 30.1–30.3 | 33.6–34.2 | 0.590 |
| | Non-pregnant | 135 | 32.0 | 31.9 | 30.4–34.1 | 30.1–30.5 | 33.7–34.1 | |
| RDW-CV | Pregnant | 381 | 14.9 | 14.7 | 12.5–16.1 | 12.4–12.6 | 15.9–16.7 | 0.0001* |
| | Non-pregnant | 127 | 12.3 | 12.4 | 12.1–14.7 | 12.0–12.4 | 14.5–15.0 | |
| RDW-SD | Pregnant | 389 | 49.0 | 48.3 | 42.1–58.2 | 41.1–42.1 | 57.3–60.9 | 0.001* |
| | Non-pregnant | 133 | 45.5 | 45.4 | 39.7–57.3 | 38.4–41.1 | 55.5–58.2 | |
| Platelet (x10⁹/L) | Pregnant | 393 | 256 | 253 | 132–373 | 125–155 | 369–403 | 0.0001* |
| | Non-pregnant | 136 | 310 | 304 | 173–456 | 171–206 | 429–463 | |
| MPV (fL) | Pregnant | 395 | 8.8 | 8.7 | 7.2–10.2 | 7.0–7.6 | 10.0–10.4 | 0.024* |
| | Non-pregnant | 136 | 8.6 | 8.5 | 7.1–10.1 | 6.8–7.3 | 10.1–10.8 | |
| PDW | Pregnant | 393 | 15.8 | 15.8 | 15.2–16.4 | 15.1–15.3 | 16.4–16.6 | 0.0001* |
| | Non-pregnant | 133 | 14.5 | 14.0 | 14.1–15.5 | 14.0–14.4 | 15.1–16.1 | |
| PCT | Pregnant | 384 | 0.216 | 0.217 | 0.121–0.316 | 0.110–0.136 | 0.306–0.336 | 0.0001* |
| | Non-pregnant | 136 | 0.269 | 0.267 | 0.168–0.382 | 0.161–0.193 | 0.349–0.390 | |

LLC-Lower limit confidence interval, ULC- Upper limit confidence interval, WBC-white blood cell count, Lymph#—Absolute lymphocyte count, MID#—Absolute mixed cell count, Gran# -Absolute granulocyte count, Lymph%—lymphocyte percentage, MID%—mixed cell percentage, Gran%—granulocyte percentage, Hgb-hemoglobin, RBC- red blood cell count, HCT- hematocrit, MCV- mean cell volume, MCH- mean cell hemoglobin, MCHC- mean cell hemoglobin concentration, RDW-CV- red cell distribution width coefficient of variation, RDW-SD- red cell distribution width standard deviation, MPV- mean platelet volume, PDW- platelet distribution width, PCT- Plateletcrit.

Mann-Whitney U-test for non-normally distributed parameters was done between pregnant and non-pregnant women.

P < 0.05 was considered as statistically significant.

*Statistically significant.

Note: The number of reference subjects varied for each parameter because of exclusion of variable number of outliers.

**Table 3. Mean, median, 95%(2.5th-97.5th) RIs with 90% CI for lower and upper reference limit of hematological parameters stratified by gestational period (trimester) of South Wollo zone, Amhara Region, Northeast Ethiopia, from June to August, 2019.**

| Parameter | Trimester | N | Mean | Median | RI (2.5th-97.5th) | 90% CI | | P-value | | | |
|---|---|---|---|---|---|---|---|---|---|---|---|
| | | | | | | LLC | ULC | 1st & 2nd | 1st & 3rd | 2nd & 3rd | 1st, 2nd & 3rd |
| WBC (x10^9/L) | 1st | 135 | 7.9 | 7.7 | 3.6–13.2 | 3.3–4.3 | 12.1–15.4 | 0.019* | 0.006* | 0.724 | 0.001* |
| | 2nd | 130 | 8.5 | 8.2 | 4.56–13.59 | 4.0–5.0 | 13.1–15.2 | | | | |
| | 3rd | 131 | 8.7 | 8.6 | 4.56–13.62 | 3.5–5.1 | 12.2–14.1 | | | | |
| Lymph# (x10^9/L) | 1st | 135 | 2.0 | 2.0 | 1.1–2.8 | 0.9–1.2 | 2.7–2.9 | 0.004* | 0.020* | 0.562 | 0.001* |
| | 2nd | 130 | 1.79 | 1.8 | 1.03–2.6 | 0.7–1.2 | 2.6–2.8 | | | | |
| | 3rd | 130 | 1.92 | 1.9 | 1.13–2.77 | 1.0–1.3 | 2.7–3.0 | | | | |
| MID# (x10^9/L) | 1st | 132 | 0.50 | 0.5 | 0.2–0.9 | 0.2–0.3 | 0.8–0.9 | 0.066 | 0.059 | 0.923 | 0.198 |
| | 2nd | 129 | 0.56 | 0.5 | 0.2–1.08 | 0.2–0.3 | 0.9–1.1 | | | | |
| | 3rd | 129 | 0.57 | 0.5 | 0.2–1.08 | 0.2–0.3 | 1.0–1.1 | | | | |
| GRAN# (x10^9/L) | 1st | 130 | 5.5 | 5.3 | 2.23–8.62 | 2.1–2.4 | 7.8–10.2 | 0.002* | 0.004* | 0.012* | 0.002* |
| | 2nd | 128 | 6.1 | 5.9 | 2.42–9.78 | 2.2–2.7 | 9.5–10.6 | | | | |
| | 3rd | 131 | 6.3 | 6.2 | 2.61–10.23 | 2.3–2.9 | 9.6–10.8 | | | | |
| Lymph (%) | 1st | 135 | 26.48 | 24.9 | 12.78–45.60 | 11.4–14.0 | 42.1–49.8 | 0.001* | 0.002* | 0.106 | 0.001* |
| | 2nd | 126 | 21.41 | 20.8 | 10.96–32.96 | 9.0–13.4 | 30.5–35.4 | | | | |
| | 3rd | 128 | 23.03 | 22.2 | 13.53–45.68 | 12.3–14.9 | 44.3–48.1 | | | | |
| MID (%) | 1st | 135 | 6.8 | 6.8 | 3.94–12.00 | 2.8–4.5 | 11.9–12.9 | 0.451 | 0.990 | 0.548 | 0.769 |
| | 2nd | 126 | 6.7 | 6.65 | 3.74–9.60 | 3.0–4.4 | 9.4–9.7 | | | | |
| | 3rd | 130 | 7.0 | 6.8 | 3.83–12.33 | 3.2–4.4 | 10.8–12.7 | | | | |
| GRAN (%) | 1st | 135 | 67.8 | 65.8 | 45.86–80.42 | 44.3–48.7 | 80.1–82.2 | 0.001* | 0.007* | 0.003* | 0.001* |
| | 2nd | 127 | 71.7 | 70.4 | 58.62–81.50 | 55.0–61.2 | 81.3–84.9 | | | | |
| | 3rd | 131 | 72.1 | 71.9 | 60.53–82.55 | 58.7–62.1 | 81.6–85.3 | | | | |
| HGB (g/dL) | 1st | 136 | 11.1 | 12.3 | 10.37–13.53 | 10.0–10.8 | 13.2–14.4 | 0.005* | 0.050* | 0.006* | 0.016* |
| | 2nd | 130 | 10.4 | 11.6 | 9.99–12.90 | 9.5–10.4 | 12.3–13.2 | | | | |
| | 3rd | 131 | 11.9 | 12.7 | 10.68–13.71 | 10.3–10.9 | 13.34–14.1 | | | | |
| RBC (x10^12/L) | 1st | 135 | 3.94 | 3.86 | 3.58–4.90 | 3.52–3.68 | 4.35–5.01 | 0.001* | 0.004* | 0.009* | 0.001* |
| | 2nd | 128 | 3.45 | 3.49 | 3.35–4.01 | 3.31–3.40 | 3.87–4.31 | | | | |
| | 3rd | 130 | 3.79 | 3.78 | 3.76–4.99 | 3.32–3.98 | 4.89–5.15 | | | | |
| HCT (%) | 1st | 136 | 41.71 | 41.05 | 34.86–47.80 | 31.6–35.7 | 46.1–49.7 | 0.007* | 0.048* | 0.008* | 0.024* |
| | 2nd | 129 | 40.56 | 40.3 | 33.93–46.19 | 32.5–34.9 | 45.3–49.6 | | | | |
| | 3rd | 131 | 39.42 | 39.9 | 32.33–45.98 | 30.9–33.6 | 45.3–47.8 | | | | |
| MCV (fL) | 1st | 133 | 94.55 | 94.2 | 86.67–103.03 | 84.7–89.3 | 100.9–104.8 | 0.067 | 0.030* | 0.927 | 0.02* |
| | 2nd | 126 | 94.72 | 94.4 | 86.10–103.58 | 83.9–88.8 | 102.5–106.6 | | | | |
| | 3rd | 129 | 95.96 | 95.0 | 87.62–105.77 | 84.0–89.8 | 103.0–107.8 | | | | |
| MCH (pg) | 1st | 130 | 29.94 | 30.0 | 26.40–32.94 | 26.3–27.2 | 32.3–33.8 | 0.121 | 0.204 | 0.167 | 0.055 |
| | 2nd | 124 | 30.20 | 30.1 | 26.89–33.20 | 26.4–28.0 | 33.0–34.3 | | | | |
| | 3rd | 124 | 30.51 | 30.4 | 27.51–33.99 | 26.8–28.0 | 33.6–34.4 | | | | |
| MCHC (g/dL) | 1st | 135 | 32.16 | 32.5 | 30.30–33.66 | 29.7–30.8 | 33.7–34.6 | 0.321 | 0.559 | 0.270 | 0.234 |
| | 2nd | 124 | 31.86 | 31.9 | 30.13–33.2 | 30.0–30.5 | 33.1–33.5 | | | | |
| | 3rd | 130 | 31.69 | 32.6 | 30.31–33.86 | 29.9–30.5 | 33.2–34.1 | | | | |
| RDW-CV | 1st | 134 | 14.07 | 14.1 | 12.44–15.99 | 12.1–12.5 | 15.7–16.6 | 0.075 | 0.458 | 0.140 | 0.064 |
| | 2nd | 127 | 14.03 | 14.2 | 12.52–17.00 | 12.0–12.7 | 16.2–17.2 | | | | |
| | 3rd | 125 | 14.05 | 13.9 | 12.62–16.20 | 12.5–12.9 | 15.9–16.7 | | | | |

(*Continued*)

**Table 3.** (Continued)

| Parameter | Trimester | N | Mean | Median | RI (2.5th-97.5th) | 90% CI | | P-value | | | |
|---|---|---|---|---|---|---|---|---|---|---|---|
| | | | | | | LLC | ULC | 1st& 2nd | 1st& 3rd | 2nd& 3rd | 1st, 2nd& 3rd |
| RDW-SD | 1st | 134 | 47.49 | 46.5 | 40.60–55.50 | 38.4–42.1 | 54.6–56.4 | 0.120 | 0.071 | 0.894 | 0.23 |
| | 2nd | 128 | 49.78 | 49.6 | 42.28–57.99 | 41.1–43.9 | 56.4–61.9 | | | | |
| | 3rd | 127 | 49.66 | 49.2 | 41.30–59.84 | 40.3–43.9 | 58.2–60.0 | | | | |
| Platelet (x10^9/L) | 1st | 133 | 258.5 | 273 | 167.05–390.00 | 155–182 | 369–400 | 0.005* | 0.001* | 0.001* | 0.005* |
| | 2nd | 130 | 253.3 | 253 | 149.58–373.32 | 140–164 | 365–390 | | | | |
| | 3rd | 131 | 251.6 | 245 | 124.60–356.90 | 90–132 | 369–386 | | | | |
| MPV (fL) | 1st | 136 | 8.25 | 8.6 | 6.73–9.80 | 6.3–7.1 | 9.4–10.3 | 0.003* | 0.001* | 0.677 | 0.001* |
| | 2nd | 129 | 8.54 | 8.78 | 7.05–10.25 | 7.1–7.4 | 9.9–10.4 | | | | |
| | 3rd | 130 | 8.71 | 8.85 | 7.40–10.30 | 7.6–7.8 | 10.0–10.5 | | | | |
| PDW | 1st | 136 | 15.74 | 15.7 | 15.10–16.36 | 15.1–15.3 | 16.2–16.4 | 0.624 | 0.000* | 0.000* | 0.001* |
| | 2nd | 128 | 15.77 | 15.8 | 15.22–16.48 | 15.0–15.3 | 16.4–16.5 | | | | |
| | 3rd | 130 | 15.92 | 15.9 | 15.16–16.57 | 15.1–15.3 | 16.5–16.8 | | | | |
| PCT | 1st | 131 | 0.225 | 0.22 | 0.152–0.316 | 0.113–0.161 | 0.305–0.337 | 0.161 | 0.209 | 0.351 | 0.054 |
| | 2nd | 127 | 0.216 | 0.217 | 0.110–0.321 | 0.108–0.132 | 0.309–0.335 | | | | |
| | 3rd | 127 | 0.209 | 0.21 | 0.118–0.321 | 0.090–0.136 | 0.291–0.333 | | | | |

LLC-Lower limit confidence interval, ULC- Upper limit confidence interval.

WBC-white blood cell count, Lymph#- Absolute lymphocyte count, MID#- Absolute mixed cell count, Gran#-Absolute granulocyte count, Lymph%- lymphocyte percentage, MID%- mixed cell percentage, Gran%- granulocyte percentage, Hgb- hemoglobin, RBC- red blood cell count, HCT- hematocrit, MCV- mean cell volume, MCH- mean cell hemoglobin, MCHC- mean cell hemoglobin concentration, RDW-CV- red cell distribution width coefficient of variation, RDW-SD- red cell distribution width standard deviation, MPV- mean platelet volume, PDW- platelet distribution width, PCT- Plateletcrit.

Mann-Whitney U-test for non-normally distributed parameters was done between trimesters (1st Vs 2nd, 1st Vs 3rd, and 2nd Vs 3rd) and Krusal-Wallis test for non-normally distributed parameters was done between inter-trimesters (1st Vs 2nd Vs 3rd).

P < 0.05 was considered as statistically significant.

* **S**tatistically significant.

Note: The number of reference subjects varied for each parameter because of the variable number of outliers excluded.

trimesters, MPV and PDW exhibited statistically significant differences. The median platelet count fell as the gestational period progressed. MPV and PDW, on the other hand, increased as the gestational period progressed (Table 3).

## Proportion of out of range values by comparing established RIs of pregnant women with currently utilized manufacturer provided RIs

Table 4 depicts the proportions of out of range (OOR) values in the pregnant women by comparing established hematological parameters RIs with currently utilized manufacturer provided RIs. The result showed that 19.1%, 19.3%, 23.8%, 26.4%, 27.8%, 43.3% and 44.9% out of range values were observed for WBC count, PLT count, Hgb, Gran#, Lymph%, Gran% and MCHC, respectively. The lowest proportion of out of range value was found in MID# (1.3%) and RDW-CV (1.6%).

## Discussion

Despite the fact that pregnancy causes changes in normal laboratory values [35–37] and that several factors, including sex, affect RIs, as demonstrated by several studies [7,8,20,21], very

**Table 4. Proportions of out of range values in pregnant women by comparison of established hematological parameters RIs with currently utilized manufacturer provided RIs.**

| Parameters | Manufacturer RIs | Currently established RIs | Out of range | | | |
| --- | --- | --- | --- | --- | --- | --- |
| | | | Lower limit (N) | Upper limit (N) | Total | |
| | | | | | N | % |
| WBC(x10$^9$/L) | 4.0–10.0 | 4.0–13.2 | 0 | 86 | 86 | 19.1 |
| Lymph#(x10$^9$/L) | 0.8–4.0 | 1.1–2.7 | 5 | 11 | 16 | 3.6 |
| MID#(x10$^9$/L) | 0.1–1.5 | 0.2–1.0 | 0 | 6 | 6 | 1.3 |
| GRAN#(x10$^9$/L) | 2.0–7.0 | 2.2–9.8 | 3 | 104 | 107 | 23.8 |
| Lymph (%) | 20.0–40.0 | 12.9–38.1 | 120 | 5 | 125 | 27.8 |
| MID (%) | 3.0–15.0 | 3.9–10.9 | 6 | 10 | 16 | 3.6 |
| GRAN (%) | 50.0–70 | 50.5–81.5 | 1 | 194 | 195 | 43.3 |
| HGB(g/dL) | 11.0–15.0 | 10.1–13.7 | 22 | 97 | 119 | 26.4 |
| RBC(x10$^{12}$/L) | 3.50–5.00 | 3.45–4.67 | 3 | 56 | 56 | 12.4 |
| HCT (%) | 37.0–47.0 | 33.5–46.5 | 53 | 3 | 55 | 12.2 |
| MCV(fL) | 80.0–100.0 | 84.8–103.5 | 9 | 40 | 49 | 10.9 |
| MCH(pg) | 27.0–34.0 | 27.5–33.0 | 4 | 8 | 12 | 2.7 |
| MCHC(g/dL) | 32.0–36.0 | 30.3–33.7 | 190 | 12 | 202 | 44.9 |
| RDW-CV | 11.0–16.0 | 12.5–16.1 | 7 | 0 | 7 | 1.6 |
| RDW-SD | 35.0–56.0 | 42.1–58.2 | 8 | 15 | 23 | 5.1 |
| Platelet(x10$^9$/L) | 100–300 | 131.7–373.2 | 9 | 78 | 87 | 19.3 |
| MPV(fL) | 6.5–12.0 | 7.1–10.1 | 9 | 2 | 11 | 2.4 |
| PDW | 9.0–17.0 | 15.2–16.4 | 8 | 10 | 18 | 4.0 |
| PCT | 0.108-.282 | 0.121–0.316 | 6 | 21 | 27 | 6.0 |

few studies in Ethiopia have been conducted to establish common hematological parameters RIs for pregnant and non-pregnant women, and none in the current study area.

The pregnant women RI derived from this study varied from those reported from Gondar [26] and Addis Ababa [38], Sudan [39], Central Uganda [40], Northwest Morocco [41], China [42] and textbooks [37] as would be expected for populations in other geographical locations with ethnic and dietary diversities. Seasonal differences in the study period, technique, and instruments used also contribute to the varied RI [8,11,37].

The lower RI of total WBC count of pregnant women in our study was lower than previously reported from different locations of Ethiopia [26,38], Central Uganda [40], and Northwest Morocco [41], as shown in a table and annexed for clarity (annex-I). The total WBC count upper RI limit in this study was greater than previous reports from Ethiopia [26,38] and Central Uganda [40], but it was equivalent to a study conducted in Northwest Morocco [41]. Moreover, the RI of total WBC count was lower than study done in United Kingdom [43]. The lower reference limit of Lymph%, MID% and Gran% among pregnant women of this study was lower than Addis Ababa [38] and Central Uganda [40], but the upper reference limit was higher than a study done in Addis Ababa, Central Uganda and Northwest Morocco [38,40,41]. These differences are postulated to occur due to factors such as genetics, dietary patterns, ethnic origin and environmental exposure to endemic pathogens which are known to influence these parameters [8,11,30,37].

In the current study, there was a statistically significant difference between pregnant and non-pregnant women in which pregnant women had a higher mean value of total WBC count than non-pregnant women, confirming results of other studies from India [19], North Morocco [37], Nigeria [44] and Sudan [39]. This is due to the physiological stress induced by the pregnancy state [45]. According to gestational age, the mean value of total WBC count

showed parallel trend with a study conducted in Addis Ababa [38] and Central Uganda [40], with a slight increase as gestation period increased. This may be because of a complex physiological process (serial endocrine system and metabolic changes) that increases WBC by accepting stimulatory signals as pregnancy advances [46]. There were statistically significant differences between pregnant and non-pregnant women in all WBC parameters, which are consistent with other findings [39,41,47]. Pregnant women had higher mean values in Gran # and Gran %, than the non-pregnant counterpart likely due to impaired neutrophilic apoptosis in pregnancy. Neutrophil cytoplasm shows toxic granulation. Neutrophil chemotaxis and phagocytic activities are depressed, especially due to the inhibitory factors present in the serum of pregnant women [48].

On the other hand, the reduction of lymphocyte levels in a pregnant woman is a natural consequence of conception and it is a normal body process. It is quite natural for the immune system to observe the embryo as something harmful and reject it. Therefore, the body ends up suppressing the immune system's response by cutting down the lymphocyte count, which allows the embryo to implant successfully and grow into a fetus. Even during this stage, the body still keeps the mother protected. Other entities such as granulocytes are activated, which temporarily take on the duties of protecting the body from external attacks [45,46].

In the current study, RIs of Hgb (10.1–13.7g/dL) and HCT (33.49–45.98%) in the pregnant women was higher than a study done in Lagos (Hgb = 9.08–12.80 g/dL; HCT = 24.61–35.71%) [49]. In the contrary, the RIs of Hgb in the present study was lower than RI determined earlier in Addis Ababa (Hgb = 13.3–14.7 g/dL) [38]. The varying RIs observed in the above results are most likely due to altitude differences, as the current study's altitude (2132 meters above sea level) was much higher than Lagos (41m above sea level), but lower than Addis Ababa (2355 meters above sea level). These increases appear to be the result of both increased erythropoiesis, which is secondary to the hypoxic stimulus, and the decreased in plasma volume that occurs at high altitude. Seasonal variations, diet, ethnic background, method and instrument used for analysis may also play a role in the variations [8,12,11, and 37].

The finding of lower RBC count, HCT and Hgb in pregnant than non-pregnant women was expected and is in line with other studies [40,41,47]. In normal pregnancy, there is an increase in erythropoietic activity but at the same time an increase in plasma volume of 40% to 50% over the non-pregnant state. This increase in plasma volume is more as compared to red cell mass leading to "hemodilution" which creates a sort of artificial anemia called "the physiological anemia of pregnancy" [8,35,50].

The mean and RIs of RBC count and Hgb were lower in the 2nd trimester compared to 1st and 3rd trimesters and similar trend was observed with a study conducted in Sudan [39]. Despite that the mean value of HCT decreased as gestational age increases and this finding is consistent with other studies [26,47,51]. The high Hgb levels in the first trimester are subsequently lowered by hemodilution in the second trimester while compensatory mechanisms (maternal plasma volume reduction and increased atrial-natriuretic peptides) raise Hgb in the last trimester. The increased plasma volume, hormonal changes and conditions that promote fluid retention contribute for reduction of HCT values as gestational period advances [35].

The pregnant women RIs of MCV in the current study was higher than study conducted in different countries [38,41,52]. The upper and lower limit of MCH RIs in this study was lower than a study reported from Gondar [26] and Addis Ababa [38], but higher than central Uganda [40]. In addition, the upper and lower limit of MCHC RI in this study was higher than central Uganda [40], and lower than Addis Ababa [38] and Northwest Morocco [41], China [28]. These discrepancies might be due to Altitude, nutritional factor, method and instrument used [12,37].

Pregnant women had higher mean values of MCV, MCH, RDW-CV and RDW-SD than their counterparts. The mean value of MCV increased as the gestational period advances,

which is consistent with another from Gondar [26]. This might be due to increased production of RBCs to meet pregnancy demands, leading to a higher proportion of young RBCs outpouring from the hemopoietic organ, which are large in size [14].

The lower limit of platelet count RIs in the current study is lower than the limits reported from Gondar [26], Addis Ababa [38] and Central Uganda [40]. Furthermore, the upper and lower RI limit of this study is lower than the study from United Kingdom [53]. However, the upper limit of platelet count RIs in the present study is higher than in other areas [26,38,40]. Similarly, the upper and lower limit RI of MPV is higher than a study done in central Uganda [40], but lower than in Northwest Morocco [41]. Among the plausible explanations for these findings include the variation of geographical location, genetic background and nutritional factors [54,55].

The mean value of platelet count and PCT were higher in non-pregnant women than in pregnant women. Dilution of platelets by the increased plasma volume that occurs during pregnancy is an apparent mechanism. The 50% increase in spleen size that occurs during pregnancy would also contribute to a lower platelet count [56,57]. The decrease in platelet count with gestational age is consistent with studies conducted in Gondar [26], Sudan [39], Central Uganda [40] and China [42]. This may be due to the continuous expansion of the uterine wall during pregnancy to accommodate fetal growth. This causes laceration of blood vessels in the uterus, leading to massive hemorrhage. In addition, hemodilution [14,35], increased consumption of platelets as well as decreased life span in the uteroplacental circulation have been suggested to be the explanation for the reduction in the number of circulating platelets during pregnancy [58].

The RIs established in the proposed investigation for non-pregnant women deviated from those established in other countries, as shown in annex-II. The current study's upper and lower RI limits for WBC were higher than those found in earlier studies from Amhara region [24], Northwest Morocco [59], and Nigeria (RI) [21] but comparable to those found in Dire Dawa [60]. In this study, the lower RI limit of Hgb was higher than the lower RI limit of studies done in Dire Dawa [60], Amhara [24] and Northwest Morocco [59], whereas the upper RI limit was lower than other studies [24,61]. The lower RI limit of RBC in the current study was higher than in previous studies in Dire Dawa [60], Amhara [24], Asmara [61], Northwest Morocco [59], Barcelona [62], and textbook [63], but it was comparable with Nigeria [21]. Furthermore, the upper RI limit of RBC in the current study was lower than in previous studies in Amhara [24], Asmara [61], and Northwest Morocco [59], Barcelona [62], and textbooks [63]. The platelet count, HCT, and MCV RI determined in this study were higher than previous studies in Northwest Morocco [59] and Barcelona [62]. Seasonal variations, genetic diversity, nutritional factors, altitude, study design, and the instrument used could all account for the observed discrepancies [8,12,37,41].

In the present study, the proportion of out of range values were 19.1%, 19.3%, 23.8%, 26.4%, 27.8%, 43.3%, and 44.9% in WBC count, PLT count, Hgb, Gran#, Lymph%, Gran%, and MCHC, respectively. The lowest proportion of out of range value was found in MID# (1.3%) and RDW-CV (1.6%). Misclassification of pregnant women in the value of WBC and Hgb is consistent with earlier study in China [28]. This could increase the risk of overlooking important physiologic alterations resulting from pathological conditions. In addition, it may increase unnecessary therapeutic actions without determining the real cause of the abnormality.

Although this study meets the minimum CLSI requirements for establishing valid RIs, few limitations existed such as lack of screening of multiple pregnancies due to resource limitations which may influence the results to some extent.

## Conclusion and recommendation

The current study is the first attempt to determine hematological parameters RIs in apparently healthy pregnant and non-pregnant women living in the South Wollo Zone, Amhara region, Northeast Ethiopia. The findings suggest the need for establishment of local reference intervals for the optimal management of maternal and fetal medical care. The results of this analysis could also be useful for developing and updating guidelines, as well as interpreting laboratory data in clinical trials. We recommend patient management and interpretation of laboratory findings of pregnant and non- pregnant women should be based on the locally derived hematological parameters RIs and suggested to be utilized by all health facilities of south wollo zone. Similarly, we recommend that the RIs for the majority of hematological parameters be updated to account for the gestational period. Conducting similar nationwide study to determine the hematological parameter RIs of pregnant women in Ethiopian population as a whole is warranted.

**Annex I. Comparison of hematological parameters RIs for pregnant women with gestational age (trimester) form different countries.**

| Parameters | Trimester | Current study | Gondar, Ethiopia(26) | Addis Ababa, Ethiopia (38) | Sudan (39) | Central Uganda (40) | Northwest morocco (41) | China (42) | Textbook (predominantly from Europeans ) (36) |
|---|---|---|---|---|---|---|---|---|---|
| WBC (x10⁹/L) | 1st | 3.64–13.24 | 8.62–10.30 | 6.5–8 | 4.36–11.20 | 4.57–8.75 | 4.5–11.6 | 4.3–12.2 | 5.7–13.6 |
| | 2nd | 4.56–13.59 | 8.67–9.90 | 8–8.7 | 5.48–12.13 | 4.71–8.85 | 4.6–12.6 | 5.1–13.8 | 6.2–14.8 |
| | 3rd | 4.56–13.62 | 8.60–9.61 | 8–8.8 | 5.00–11.96 | 4.8–9.2 | 5.3–14.3 | 4.9–13.4 | 5.9–16.9 |
| | Combined | 4.00–13.21 | 8.90–9.60 | 8.1–8.6 | | 4.51–8.79 | 4.6–13.0 | | |
| Lymph# (x10⁹/L) | 1st | 1.1–2.8 | 2.18–2.53 | | 1.20–2.98 | 1.30–2.58 | 1.2–3.14 | | 1.1–3.5 |
| | 2nd | 1.03–2.6 | 1.96–2.22 | | 1.28–2.63 | 1.13–2.45 | 1.2–3.6 | | 0.9–3.9 |
| | 3rd | 1.13–2.77 | 2.11–2.33 | | 1.10–2.60 | 0.84–2.02 | 1.1–3.8 | | 1–3.6 |
| | Combined | 1.1–2.71 | 2.13–2.28 | | | 1.08–2.38 | 1.2–3.6 | | |
| MID# (x10⁹/L) | 1st | 0.2–0.9 | | | 0.30–0.99 | 0.29–0.73 | | | |
| | 2nd | 0.2–1.08 | | | 0.30–0.90 | 0.28–0.76 | | | |
| | 3rd | 0.2–1.08 | | | 0.30–1.00 | 0.31–0.77 | | | |
| | Combined | 0.2–1.0 | | | | 0.29–0.73 | | | |
| GRAN# (x10⁹/L) | 1st | 2.23–8.62 | | | 2.56–8.68 | 3.55–5.83 | 2.1–8.2 | | 3.6–10.1 |
| | 2nd | 2.42–9.78 | | | 3.62–9.80 | 2.59–6.13 | 2.2–9.2 | | 3.8–12.3 |
| | 3rd | 2.61–10.23 | | | 3.22–9.30 | 2.7–5.68 | 3–11 | | 3.9–13.1 |
| | Combined | 2.2–9.83 | | | | 2.48–5.86 | 2.2–9.7 | | |
| Lymph (%) | 1st | 12.78–45.60 | | 23.4–30 | | 20.91–39.83 | | | |
| | 2nd | 10.96–32.96 | | 21.6–23.4 | | 19.96–36.8 | | | |
| | 3rd | 13.53–45.68 | | 23–25 | | 15.93–32.89 | | | |
| | Combined | 12.9–38.13 | | 23–24.3 | | 19.53–36.91 | | | |
| MID (%) | 1st | 3.94–12.00 | | 7–12 | | 5.72–9.68 | | | |
| | 2nd | 3.74–9.60 | | 8–9 | | 5.11–10.45 | | | |
| | 3rd | 3.83–12.33 | | 8–10 | | 6.64–10.84 | | | |
| | Combined | 3.9–10.9 | | 8.3–9.3 | | 5.78–10.3 | | | |
| GRAN (%) | 1st | 45.86–80.42 | | 60–68 | | | | | |
| | 2nd | 58.62–81.50 | | 68–70 | | | | | |
| | 3rd | 60.53–82.55 | | 65.6–68 | | | | | |
| | Combined | 50.45–81.54 | | 67–68.7 | | | | | |

*(Continued)*

**Annex I.** (*Continued*)

| Parameters | Trimester | Current study | Gondar, Ethiopia(26) | Addis Ababa, Ethiopia (38) | Sudan (39) | Central Uganda (40) | Northwest morocco (41) | China (42) | Textbook (predominantly from Europeans) (36) |
|---|---|---|---|---|---|---|---|---|---|
| HGB (g/dL) | 1st | 10.37–13.53 | 12.43–13.46 | 13.7–15 | 8.92–12.74 | 11.6–13.32 | 10–13.9 | 10.4–14.0 | 11.0–14.3 |
| | 2nd | 9.99–12.90 | 12.82–13.33 | 12.6–16 | 9.00–12.10 | 10.71–12.29 | 9.6–13.6 | 9.5–13.0 | 10.0–13.7 |
| | 3rd | 10.68–13.71 | 13.11–13.67 | 12.6–14 | 8.82–12.60 | 10.85–12.65 | 9.1–13.4 | 9.6–13.5 | 9.8–13.7 |
| | Combined | 10.1–13.67 | 12.99–13.36 | 13.3–14.7 | | 10.79–12.79 | 9.4–13.7 | | |
| RBC (x10^12/L) | 1st | 3.58–4.90 | 4.08–4.46 | 4.6–5 | 3.69–4.93 | 4.04–5.02 | 3.49–4.91 | 3.35–4.75 | 3.52–4.52 |
| | 2nd | 3.35–4.01 | 4.21–4.43 | 4.4–4.5 | 3.69–4.93 | 3.77–4.67 | 3.26–4.82 | 3.01–4.31 | 3.2–4.41 |
| | 3rd | 3.76–4.99 | 4.37–4.55 | 4.4–4.6 | 3.44–4.78 | 3.92–4.7 | 3.19–4.78 | 3.08–4.50 | 3.1–4.44 |
| | Combined | 3.45–4.67 | 4.30–4.44 | 4.4–4.5 | | 3.86–4.84 | 3.29–4.85 | | |
| HCT (%) | 1st | 34.86–47.80 | 37.17–41.19 | 40–43 | 30.12–40.30 | 36.94–45.3 | 29.8–40.9 | 31–41 | 31–41 |
| | 2nd | 33.93–46.19 | 39.63–41.44 | 38–40 | 30.58–38.23 | 34.67–41.35 | 28.6–39.9 | 30–39 | 30–38 |
| | 3rd | 32.33–45.98 | 41.17–42.75 | 39–40 | 29.66–40.04 | 32.47–44.79 | 27.34–39.3 | 30–41 | 28–39 |
| | Combined | 33.49–46.52 | 40.19–41.49 | 39–39.9 | | 34.84–43.66 | 28.6–40.5 | | |
| MCV (fL) | 1st | 86.67–103.03 | 92.02–94.34 | 85–88 | 65.50–93.02 | 83.06–98.89 | 74.4–94.9 | 82.3–98.2 | 81–96 |
| | 2nd | 86.10–103.58 | 93.18–95.20 | 86.7–88.4 | 71.35–94.70 | 83.71–97.09 | 74.7–97.7 | 84.2–102.7 | 82–97 |
| | 3rd | 87.62–105.77 | 93.09–95.37 | 88–89 | 73.40–95.68 | 81.69–97.43 | 72.8–96.1 | 82.4–103.9 | 81–99 |
| | Combined | 84.76–103.52 | 93.33–94.63 | 87.4–88.5 | | 82.52–97.16 | 74–96 | | |
| MCH (pg) | 1st | 26.40–32.94 | 29.82–31.14 | 29.4–30.5 | 19.40–28.74 | 24.96–30 | 24.2–32.9 | 27.8–33.4 | |
| | 2nd | 26.89–33.20 | 29.84–31.11 | 30–30.6 | 20.83–30.15 | 24.86–29.94 | 24.0–33.3 | 27.6–34.7 | |
| | 3rd | 27.51–33.99 | 26.73–40.79 | 30.4–31 | 21.34–30.18 | 23.95–28.81 | 23–33.4 | 27.0–35.0 | |
| | Combined | 27.5–33.00 | 28.88–34.81 | 30–30.7 | | 24.72–29.78 | 23.7–33.2 | | |
| MCHC (g/dL) | 1st | 30.30–33.66 | 37.02–33.38 | 34–35 | 28.54–32.30 | 29.46–30.98 | 31.3–36.6 | 32.2–35.4 | |
| | 2nd | 30.13–33.2 | 31.24–35.35 | 34.4–34.8 | 28.78–32.40 | 29.49–31.13 | 31.2–36.6 | 31.3–35.3 | |
| | 3rd | 30.31–33.86 | 31.75–32.41 | 34.5–35 | 28.82–32.50 | 29.37–30.93 | 30.8–36.2 | 30.9–35.1 | |
| | Combined | 30.30–33.73 | 31.91–33.37 | 34.5–34.8 | | 29.47–31.07 | 31.2–36.5 | | |
| RDW-CV | 1st | 12.44–15.99 | | | 12.00–17.94 | | | | |
| | 2nd | 12.52–17.00 | | | 12.40–18.81 | | | | |
| | 3rd | 12.62–16.20 | | | 12.11–17.50 | | | | |
| | Combined | 12.5–16.145 | | | | | | | |

(*Continued*)

**Annex I.** (Continued)

| Parameters | Trimester | Current study | Gondar, Ethiopia(26) | Addis Ababa, Ethiopia (38) | Sudan (39) | Central Uganda (40) | Northwest morocco (41) | China (42) | Textbook (predominantly from Europeans) (36) |
|---|---|---|---|---|---|---|---|---|---|
| RDW-SD | 1st | 40.60–55.50 | | | | | | | |
| | 2nd | 42.28–57.99 | | | | | | | |
| | 3rd | 41.30–59.84 | | | | | | | |
| | Combined | 42.1–58.2 | | | | | | | |
| Platelet (x10⁹/L) | 1st | 167.05–390.00 | 224.53–253.21 | 212–267 | 182.6–418.0 | 152.38–267.24 | 145–374 | 64–263 | 174–391 |
| | 2nd | 149.58–373.32 | 213.70–247.86 | 220–239 | 163.8–381.8 | 145.41–224.59 | 140–364 | 63–247 | 171–409 |
| | 3rd | 124.60–356.90 | 209.50–237.38 | 216–235 | 150.4–346.2 | 128.48–220.94 | 139–398 | 61–238 | 155–429 |
| | Combined | 131.7–373.15 | 221.25–240.14 | 221.6–235 | | 148.88–249.12 | 141–377 | | |
| MPV (fL) | 1st | 6.73–9.80 | | | 6.90–9.78 | 6.28–7.8 | 8.9–13.7 | | |
| | 2nd | 7.05–10.25 | | | 6.96–9.62 | 6.4–7.66 | 8.9–13.5 | | |
| | 3rd | 7.40–10.30 | | | 7.00–9.80 | 6.29–7.89 | 8.9–13.2 | | |
| | Combined | 7.09–10.12 | | | | 6.33–7.75 | 8.9–13.5 | | |
| PDW | 1st | 15.10–16.36 | | | 15.10–16.29 | | | | |
| | 2nd | 15.22–16.48 | | | 15.20–16.30 | | | | |
| | 3rd | 15.16–16.57 | | | 15.40–16.60 | | | | |

**Annex II.** Comparison of non-pregnant (adult female) women RIs of our study with those found in the literature.

| Hematological parameters | Current study | Dire Dawa (63) | Amhara [24] | Asmara [61] | Northwest Morocco [59] | Nigeria [21] | Barcelona [61] | Text Book [63] |
|---|---|---|---|---|---|---|---|---|
| WBC (x10⁹/L) | 3.6–10.3 | 3.8–10.2 | 3–11.2 | 3.3–8.9 | 4.1–10.7 | 4.43–4.82 | 3.9–9.5 | 3.6–10.6 |
| Lymph# (x10⁹/L) | 1.2–3.7 | 1.3–4.03 | 1.1–4.5 | | 1.2–3.8 | | 1.3–3.4 | 1.0–3.2 |
| MID# (x10⁹/L) | 0.2–0.8 | | 0.2–2.1 | | | | | |
| GRAN# (x10⁹/L) | 1.3–6.9 | 1.4–7.0 | 1.1–6.7 | | 1.8–7 | | 1.5–5.7 | 1.7–7.5 |
| Lymph (%) | 19.9–57.1 | 18.1–55.1 | 19.8–57.4 | 22.3–58.2 | | 39.0–42.1 | 21–50 | 50–70 |
| MID (%) | 3.9–10.9 | | 5.2–26.4 | | | | | |
| GRAN (%) | 33.4–71.8 | 39.9–71.2 | 27.2–71.9 | 33.5–70.5 | | 49.1–52.3 | 37.1–68.4 | 18–42 |
| HGB (g/dL) | 12.4–14.3 | 10.7–15.2 | 10.7–17.5 | 12.5–17.6 | 11–14.8 | 12.4–13.1 | 12.0–14.7 | 12.0–15.0 |
| RBC (x10¹²/L) | 4.44–5.01 | 3.81–5.49 | 3.5–5.6 | 4–5.7 | 3.86–5.2 | 4.5–5.3 | 3.9–5.1 | 3.80–5.20 |
| HCT (%) | 38.4–50.1 | 37.4–52.0 | 32.2–50.1 | 37.9–52 | 33.5–43.9 | 38.0–40.5 | 36–45 | 35–49 |
| MCV (fL) | 86.1–101.6 | 83.4–104.2 | 81–100 | 85.5–100 | 75.1–94.7 | 84.8–86.5 | 83.6–97.0 | 80–100 |
| MCH (pg) | 27.1–32.4 | 23.0–30.1 | 25.3–34.6 | 26.5–32.6 | 24–32.3 | 27.1–28.9 | 27–32 | 26–34 |
| MCHC (g/dL) | 30.4–34.1 | 26.7–29.9 | 28.8–36.9 | 30–33.7 | 31.2–36 | 31.8–32.3 | 314–319 | 32–36 |
| RDW-CV | 12.1–14.7 | 12.4–15.7 | 11.6–15.4 | 12.3–17 | | | | 11.5–14.5 |
| RDW-SD | 39.72–57.3 | 43.1–60.9 | | | | | | |
| Platelet (x10⁹/L) | 173–456 | 177–442 | 90–399 | 145.4–351.6 | 150–378 | 229.3–251.2 | 153–368 | 150–450 |
| MPV (fL) | 7.1–10.1 | 7.3–10.30 | 8–12.3 | | 9–13.7 | | 9.7–13.2 | 7.0–12.0 |

## Acknowledgments

We would like to express our deep and sincere gratitude to all participants who volunteered to take part in the study. Dessie and Kombolcha health center heads, laboratory staffs and midwives are gratefully acknowledged for their support in sample collection, processing and analysis. In addition, we would like to thank all health extension workers in the study areas for their support in recruiting subjectively well study participants.

## Author Contributions

**Conceptualization:** Mesfin Fiseha, Miftah Mohammed, Endris Ebrahim, Wondmagegn Demsiss, Mikias Negash, Zemenu Tamir, Mihret Tilahun, Aster Tsegaye.

**Data curation:** Mesfin Fiseha, Miftah Mohammed, Endris Ebrahim, Mohammed Tarekegn, Amanuel Angelo, Zemenu Tamir, Aster Tsegaye.

**Formal analysis:** Mesfin Fiseha, Miftah Mohammed, Endris Ebrahim, Amanuel Angelo, Zemenu Tamir, Aster Tsegaye.

**Funding acquisition:** Mesfin Fiseha, Wondmagegn Demsiss, Aster Tsegaye.

**Investigation:** Mesfin Fiseha, Miftah Mohammed, Wondmagegn Demsiss, Amanuel Angelo, Mihret Tilahun, Aster Tsegaye.

**Methodology:** Mesfin Fiseha, Miftah Mohammed, Endris Ebrahim, Wondmagegn Demsiss, Mohammed Tarekegn, Amanuel Angelo, Mikias Negash, Zemenu Tamir, Mihret Tilahun, Aster Tsegaye.

**Project administration:** Mesfin Fiseha, Wondmagegn Demsiss.

**Resources:** Mesfin Fiseha, Wondmagegn Demsiss, Mohammed Tarekegn, Aster Tsegaye.

**Software:** Mesfin Fiseha, Mihret Tilahun.

**Supervision:** Mesfin Fiseha, Wondmagegn Demsiss, Mohammed Tarekegn, Zemenu Tamir, Aster Tsegaye.

**Validation:** Mesfin Fiseha, Miftah Mohammed, Endris Ebrahim, Zemenu Tamir, Aster Tsegaye.

**Visualization:** Mesfin Fiseha, Miftah Mohammed, Mohammed Tarekegn, Zemenu Tamir, Mihret Tilahun, Aster Tsegaye.

**Writing – original draft:** Mesfin Fiseha, Mikias Negash, Zemenu Tamir, Mihret Tilahun, Aster Tsegaye.

**Writing – review & editing:** Mesfin Fiseha, Mikias Negash, Zemenu Tamir, Mihret Tilahun, Aster Tsegaye.

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
