## [Decision Letter · Decision Letter 0]

28 Apr 2022

PONE-D-22-08707Common Haematological Parameters Reference Intervals for Apparently Healthy Pregnant and Non-pregnant Women of South Wollo Zone, Amhara Regional State, Northeast Ethiopia.PLOS ONE

Dear Dr. Fiseha,

Thank you for submitting your manuscript to PLOS ONE. Overall, your report was well received. After review a few small suggestions have been made by one reviewer which you may wish to incorporate into a revised manuscript. 

We look forward to receiving your revised manuscript.

Kind regards,

Colin Johnson, Ph.D.

Academic Editor

PLOS ONE

Journal Requirements:

Reviewers' comments:

Reviewer's Responses to Questions

**Comments to the Author**

1. Is the manuscript technically sound, and do the data support the conclusions?

Reviewer #1: Yes

2. Has the statistical analysis been performed appropriately and rigorously? 

Reviewer #1: Yes

3. Have the authors made all data underlying the findings in their manuscript fully available?

Reviewer #1: Yes

4. Is the manuscript presented in an intelligible fashion and written in standard English?

Reviewer #1: Yes

5. Review Comments to the Author

Reviewer #1: Thank you for submitting an excellent manuscript on an important topic. This study is methodologically sound, and has been well executed with a high level of detail. I agree with the premise of the study, and the main findings.

A few points:

1) Some large reference studies on WBC and platelets are not referenced, and you may benefit from their inclusion (references below). Your reference limits differ from these in places - can you comment on why?

2) Overall, the text is verbose and should be edited for brevity before publication (particularly in the discussion and conclusion)

Dockree S, Shine B, Pavord S, Impey L, Vatish M. White blood cells in pregnancy: reference intervals for before and after delivery. EBioMedicine. 2021 Dec 1;74:103715.

Dockree S, Shine B, Impey L, Mackillop L, Randeva H, Vatish M. Improving diagnostic accuracy in pregnancy with individualised, gestational age-specific reference intervals. Clin Chim Acta. 2022 Feb 15;527:56-60. doi: 10.1016/j.cca.2022.01.007. Epub 2022 Jan 14. PMID: 35038434

6. PLOS authors have the option to publish the peer review history of their article (what does this mean?). If published, this will include your full peer review and any attached files.

Reviewer #1: **Yes: **Dr Samuel Dockree

---

## [Author Response · Author response to Decision Letter 0]

26 May 2022

Response to Reviewer and academic editor

To Academic Editor

Responses: First of all, we would like to thank you for your reminder to consider the journal formatting requirement. Therefore, we made a correction on some of the formatting issues. In addition, we have deleted two duplicate references and added a new reference as per the academic reviewer's recommendation, and a correction has been made to the reference list, which is indicated in the document.

To Reviewer: We would like to thank you very much for your pertinent comments that praised the paper quality. 

Comment 1: Some large reference studies on WBC and platelets are not referenced, and you may benefit from their inclusion (references below). Your reference limits differ from these in places - can you comment on why?

Response: Thank you. As per your recommendation, we have used the two papers and the possible explanation for the variation of the reference limit of WBC and platelet with our study is postulated to occur due to factors such as nutritional status, genetics, dietary patterns, ethnic origin, and environmental exposure to endemic pathogens.

Comment 2: Overall, the text is verbose and should be edited for brevity before publication (particularly in the discussion and conclusion)

Response: Well addressed accordingly, thanks (We tried to avoid verbose text in the document, particularly in the discussion and conclusion.)

---

## [Editor Report · Decision Letter 1]

9 Jun 2022

PONE-D-22-08707R1Common Hematological Parameters Reference Intervals for Apparently Healthy Pregnant and Non-pregnant Women of South Wollo Zone, Amhara Regional State, Northeast Ethiopia.PLOS ONE

Dear Dr. Fiseha,

Thank you for submitting your manuscript to PLOS ONE. After review, a referee has submitted a few comments and suggested references to be included. Please submit a rebuttal letter explaining any changes you may have made along with a revised manuscript.

We look forward to receiving your revised manuscript.

Kind regards,

Colin Johnson, Ph.D.

Academic Editor

PLOS ONE
---

## [Author Response · Author response to Decision Letter 1]

11 Jun 2022

Response to Reviewer and academic editor

To Academic Editor

Author Responses: First of all, we would like to thank you for your reminder to consider the journal formatting requirement. Therefore, we made a correction on some of the formatting issues. 

To Reviewer

We would like to thank you very much for your pertinent comments that praised the paper quality. 

Comment 1: Some large reference studies on WBC and platelets are not referenced, and you may benefit from their inclusion (references below). Your reference limits differ from these in places - can you comment on why?

Author Response: Thank you. As per your recommendation, we have used the two papers and the possible explanation for the variation of the reference limit of WBC and platelet with our study is postulated to occur due to factors such as nutritional status, genetics, dietary patterns, ethnic origin, and environmental exposure to endemic pathogens. As per you suggestion we have been added the following references in the manuscript as: Reference number :43. Dockree S, Shine B, Pavord S, Impey L, Vatish M. White blood cells in pregnancy: reference intervals for before and after delivery. EBioMedicine. 2021; 74:103715. Reference number: 53. Dockree S, Shine B, Impey L, Mackillop L, Randeva H, Vatish M. Improving diagnostic accuracy in pregnancy with individualised, gestational age-specific reference intervals. Clin Chim Acta. 2022; 527:56-60.

In addition, we have deleted two references because they were duplicate references, and a correction has been made to the reference list, which is indicated in the revised manuscript with tack changes document.

Comment 2: Overall, the text is verbose and should be edited for brevity before publication (particularly in the discussion and conclusion)

Author Response: Well addressed accordingly, thanks (We tried to avoid verbose text in the document, particularly in the discussion and conclusion.)

---

## [Editor Report · Decision Letter 2]

16 Jun 2022

Common Hematological Parameters Reference Intervals for Apparently Healthy Pregnant and Non-pregnant Women of South Wollo Zone, Amhara Regional State, Northeast Ethiopia.

PONE-D-22-08707R2

Dear Dr. Fiseha,

We’re pleased to inform you that your manuscript has been judged scientifically suitable for publication and will be formally accepted for publication once it meets all outstanding technical requirements.

Kind regards,

Colin Johnson, Ph.D.

Academic Editor

PLOS ONE
---

## [Editor Report · Acceptance letter]

7 Jul 2022

PONE-D-22-08707R2 

Common Hematological Parameters Reference Intervals for Apparently Healthy Pregnant and Non-pregnant Women of South Wollo Zone, Amhara Regional State, Northeast Ethiopia 

Dear Dr. Fiseha:

I'm pleased to inform you that your manuscript has been deemed suitable for publication in PLOS ONE. Congratulations! Your manuscript is now with our production department. 

Kind regards, 

on behalf of

Dr. Colin Johnson 

Academic Editor

PLOS ONE